# Deep contextualized word representations

## Abstract

We introduce a new type of *deep contextualized* word representation that models both (1) complex characteristics of word use (e.g., syntax and semantics), and (2) how these uses vary across linguistic contexts (i.e., to model polysemy). Our word vectors are learned functions of the internal states of a deep bidirectional language model (biLM), which is pretrained on a large text corpus. We show that these representations can be easily added to existing models and significantly improve the state of the art across six challenging NLP problems, including question answering, textual entailment and sentiment analysis. We also present an analysis showing that exposing the deep internals of the pretrained network is crucial, allowing downstream models to mix different types of semi-supervision signals.

## 1 Introduction

Pretrained word representations (Mikolov et al., 2013; Pennington et al., 2014) are a key component in many neural language understanding models. However, learning high quality representations can be challenging. They should ideally model both (1) complex characteristics of word use (e.g., syntax and semantics), and (2) how these uses vary across linguistic contexts (i.e., to model polysemy). In this paper, we introduce a new type of *deep contextualized* word representation that directly addresses both challenges, can be easily integrated into existing models, and significantly improves the state of the art in every considered case across a range of challenging language understanding problems.

Our representations differ from traditional word embeddings in that each word is assigned a representation that is a function of the entire input sentence. We use vectors derived from a bidirectional LSTM that is trained with a coupled language model (LM) objective on a large text corpus. For this reason, we call them ELMo (Embeddings from Language Models) representations. Unlike previous approaches for learning contextualized word vectors (Peters et al., 2017; McCann et al., 2017), ELMo representations are deep, in the sense that they are a function of all of the internal layers of the biLM. More specifically, we learn a linear combination of the vectors stacked above each input word for each end task, which markedly improves performance over just using the top LSTM layer.

Combining the internal states in this manner allows for very rich word representations. We show that, for example, the higher-level LSTM states capture context-dependent aspects of word meaning (e.g., they can be used without modification to perform well on supervised word sense disambiguation tasks) while lower-level states model aspects of syntax (e.g., they can be used to do part-of-speech tagging). Simultaneously exposing all of these signals can be highly beneficial, as models can learn to select the types of semi-supervision that are most useful for each end task.

Extensive experiments demonstrate that ELMo representations work extremely well in practice. We first show that they can be easily added to existing models for six diverse and challenging language understanding problems, including textual entailment, question answering and sentiment analysis. The addition of ELMo representations alone significantly improves the state of the art in every case, including up to 20% relative error reductions. For tasks where direct comparisons are possible, ELMo outperforms CoVe (McCann et al., 2017), which computes contextualized representations using a neural machine translation encoder. Finally, an analysis of both ELMo and CoVe reveals that deep representations outperform those derived from just the top layer of an LSTM. Our trained models and code will be made publicly available, and we expect that ELMo will provide similar gains for many other NLP problems.[1]

---

[1] http://anonymous

## 2 RELATED WORK

Due to their ability to capture syntactic and semantic information of words from large scale unlabeled text, pretrained word vectors (Turian et al., 2010; Mikolov et al., 2013; Pennington et al., 2014) are a standard component of most state-of-the-art NLP architectures, including for question answering (Wang et al., 2017), textual entailment (Chen et al., 2017) and semantic role labeling (He et al., 2017). However, these approaches for learning word vectors only allow a single, context-independent representation for each word. Another line of research focuses on global methods for learning sentence and document encoders from unlabeled data (e.g., Le & Mikolov, 2014; Kiros et al., 2015; Hill et al., 2016; Conneau et al., 2017), where the goal is to build one representation for an entire text sequence. In contrast, as we will see Section 3, ELMo representations are associated with individual words, but also encode the larger context in which they appear.

Previously-proposed methods overcome some of the shortcomings of traditional word vectors by either enriching them with subword information (e.g., Wieting et al., 2016; Bojanowski et al., 2017) or learning separate vectors for each word sense (e.g., Neelakantan et al., 2014). Our approach also benefits from subword units through the use of character convolutions, and we seamlessly incorporate multi-sense information into downstream tasks without explicitly training to predict predefined sense classes.

Other recent work has also focused on learning context-dependent representations. context2vec (Melamud et al., 2016) uses a bidirectional Long Short Term Memory (LSTM; Hochreiter & Schmidhuber, 1997) to encode the context around a pivot word. Other approaches for learning contextual embeddings include the pivot word itself in the representation and are computed with the encoder of either a supervised neural machine translation (MT) system (CoVe; McCann et al., 2017) or an unsupervised language model (Peters et al., 2017). Both of these approaches benefit from large datasets, although the MT approach is limited by the size of parallel corpora. In this paper, we take full advantage of access to plentiful monolingual data, and train our biLM on a corpus with approximately 30 million sentences (Chelba et al., 2014). We also generalize these approaches to deep contextual representations, which we show work well across a broad range of diverse NLP tasks.

Previous work has also shown that different layers of deep biRNNs encode different types of information. For example, introducing multi-task syntactic supervision (e.g., part-of-speech tags) at the lower levels of a deep LSTM can improve overall performance of higher level tasks such as dependency parsing (Hashimoto et al., 2017) or CCG super tagging (Søgaard & Goldberg, 2016). In an RNN-based encoder-decoder machine translation system, Belinkov et al. (2017) showed that the representations learned at the first layer in a 2-layer LSTM encoder are better at predicting POS tags then second layer. Finally, the top layer of an LSTM for encoding word context (Melamud et al., 2016) has been shown to learn representations of word sense. We show that similar signals are also induced by the modified language model objective of our ELMo representations, and it can be very beneficial to learn models for downstream tasks that mix these different types of semi-supervision.

Similar to computer vision where representations from deep CNNs pretrained on ImageNet are fine tuned for other tasks (Krizhevsky et al., 2012; Shelhamer et al., 2015), Dai & Le (2015) and Ramachandran et al. (2017) pretrain encoder-decoder pairs and then fine tune with task specific supervision. In contrast, after pretraining the biLM with unlabeled data, we fix the weights and add additional task-specific model capacity, allowing us to leverage large, rich and universal biLM representations for cases where downstream training data size dictates a smaller supervised model.

## 3 ELMo: EMBEDDINGS FROM LANGUAGE MODELS

This section details how we compute ELMo representations and use them to improve NLP models. We first present our biLM approach (Sec. 3.1) and then show how ELMo representations are computed on top of them (Sec. 3.2). We also describe how to add ELMo to existing neural NLP architectures (Sec. 3.3), and the details of how the biLM is pretrained (Sec. 3.4).

### 3.1 BIDIRECTIONAL LANGUAGE MODELS

Given a sequence of $N$ tokens, $(t_1, t_2, ..., t_N)$, a forward language model computes the probability of the sequence by modeling the probability of token $t_k$ given the history $(t_1, ..., t_{k-1})$:

$$p(t_1, t_2, \ldots, t_N) = \prod_{k=1}^{N} p(t_k \mid t_1, t_2, \ldots, t_{k-1}).$$

Recent state-of-the-art neural language models (Józefowicz et al., 2016; Melis et al., 2017; Merity et al., 2017) compute a context-independent token representation $\mathbf{x}_k^{LM}$ (via token embeddings or a CNN over characters) then pass it through $L$ layers of forward LSTMs. At each position $k$, each LSTM layer outputs a context-dependent representation $\overrightarrow{\mathbf{h}}_k^{LM,j}$ where $j = 1, \ldots, L$. The top layer LSTM output, $\overrightarrow{\mathbf{h}}_k^{LM,L}$, is used to predict the next token $t_{k+1}$ with a Softmax layer.

A backward LM is similar to a forward LM, except it runs over the sequence in reverse, predicting the previous token given the future context:

$$p(t_1, t_2, \ldots, t_N) = \prod_{k=1}^{N} p(t_k \mid t_{k+1}, t_{k+2}, \ldots, t_N).$$

It can be implemented in an analogous way to a forward LM, with each backward LSTM layer $j$ in a $L$ layer deep model producing representations $\overleftarrow{\mathbf{h}}_k^{LM,j}$ of $t_k$ given $(t_{k+1}, \ldots, t_N)$.

A biLM combines both a forward and backward LM. Our formulation jointly maximizes the log likelihood of the forward and backward directions:

$$\sum_{k=1}^{N} \left( \log p(t_k \mid t_1, \ldots, t_{k-1}; \Theta_x, \overrightarrow{\Theta}_{LSTM}, \Theta_s) + \log p(t_k \mid t_{k+1}, \ldots, t_N; \Theta_x, \overleftarrow{\Theta}_{LSTM}, \Theta_s) \right).$$

We tie the parameters for both the token representation ($\Theta_x$) and Softmax layer ($\Theta_s$) in the forward and backward direction while maintaining separate parameters for the LSTMs in each direction. Overall, this formulation is similar to the approach of Peters et al. (2017), with the exception that we share some weights between directions instead of using completely independent parameters. In the next section, we depart from previous work by introducing a new approach for learning word representations that are a linear combination of the biLM layers.

### 3.2 ELMO

ELMo is a task specific combination of the intermediate layer representations in the biLM. For each token $t_k$, a $L$-layer biLM computes a set of $2L + 1$ representations

$$R_k = \{\mathbf{x}_k^{LM}, \overrightarrow{\mathbf{h}}_k^{LM,j}, \overleftarrow{\mathbf{h}}_k^{LM,j} \mid j = 1, \ldots, L\} = \{\mathbf{h}_k^{LM,j} \mid j = 0, \ldots, L\},$$

where $\mathbf{h}_k^{LM,0}$ is the token layer and $\mathbf{h}_k^{LM,j} = [\overrightarrow{\mathbf{h}}_k^{LM,j}; \overleftarrow{\mathbf{h}}_k^{LM,j}]$, for each biLSTM layer.

For inclusion in a downstream model, ELMo collapses all layers in $R$ into a single vector, $\mathbf{ELMo}_k = E(R_k; \theta_e)$. In the simplest case, ELMo just selects the top layer, $E(R_k) = \mathbf{h}_k^{LM,L}$, as in TagLM (Peters et al., 2017) and CoVe (McCann et al., 2017). Across the tasks considered, the best performance was achieved by weighting all biLM layers with softmax-normalized learned scalar weights $\mathbf{s} = Softmax(\mathbf{w})$:

$$E(R_k; \mathbf{w}, \gamma) = \gamma \sum_{j=0}^{L} s_j \mathbf{h}_k^{LM,j}. \tag{1}$$

The scalar parameter $\gamma$ allows the task model to scale the entire ELMo vector and is of practical importance to aid the optimization process (see the Appendix for details). Considering that the activations of each biLM layer have a different distribution, in some cases it also helped to apply layer normalization (Ba et al., 2016) to each biLM layer before weighting.

### 3.3 Using biLMs for supervised NLP tasks

Given a pre-trained biLM and a supervised architecture for a target NLP task, it is a simple process to use the biLM to improve the task model. All of the architectures considered in this paper use RNNs, although the method is equally applicable to CNNs.

We first consider the lowest layers of the supervised model without the biLM. Most RNN based NLP models (including every model in this paper) share a common architecture at the lowest layers, allowing us to add ELMo in a consistent, unified manner. Given a sequence of tokens $(t_1, \ldots, t_N)$, it is standard to form a context-independent token representation $\mathbf{x}_k$ for each token position using pre-trained word embeddings and optionally character-based representations (typically from a CNN). Then, one or more layers of bidirectional RNNs compute a context-sensitive representation $\mathbf{h}_k$ for each token position $k$, where $\mathbf{h}_k$ is the concatenation $[\overrightarrow{\mathbf{h}}_k; \overleftarrow{\mathbf{h}}_k]$ of the forward and backward RNNs.

To add ELMo to the supervised model, we first freeze the weights of the biLM and then concatenate the ELMo vector $\mathbf{ELMo}_k$ with $\mathbf{x}_k$ and pass the ELMo enhanced representation $[\mathbf{x}_k; \mathbf{ELMo}_k]$ into the task RNN. For some tasks (e.g., SNLI, SQuAD), we observe further improvements by also including ELMo at the output of the task RNN by replacing $\mathbf{h}_k$ with $[\mathbf{h}_k; \mathbf{ELMo}_k]$. As the remainder of the supervised model remains unchanged, these additions can happen within the context of more complex neural models. For example, see the SNLI experiments in Sec. 4 where a bi-attention layer follows the biLSTMs, or the coreference resolution experiments where a clustering model is layered on top of the biLSTMs that compute embeddings for text spans.

Finally, we found it beneficial to add a moderate amount of dropout to ELMo (Srivastava et al., 2014) and in some cases to regularize the ELMo weights by adding $\lambda \|\mathbf{w} - \frac{1}{L+1}\|_2^2$ to the loss. This regularization term imposes an inductive bias on the ELMo weights to stay close to an average of all biLM layers.

### 3.4 Pre-trained bidirectional language model architecture

The pre-trained biLMs in this paper are similar to the architectures in Józefowicz et al. (2016) and Kim et al. (2015), but modified to support joint training of both directions and to include a residual connection between LSTM layers. We focus on biLMs trained at large scale in this work, as Peters et al. (2017) highlighted the importance of using biLMs over forward-only LMs and large scale training. To balance overall language model perplexity with model size and computational requirements for downstream tasks while maintaining a purely character-based input representation, we halved all embedding and hidden dimensions from the single best model `CNN-BIG-LSTM` in (Józefowicz et al., 2016). The resulting model uses 2048 character n-gram convolutional filters followed by two highway layers (Srivastava et al., 2015) and a linear projection down to a 512 dimension token representation. Each recurrent direction uses two LSTM layers with 4096 units and 512 dimension projections. The average forward and backward perplexities on the 1B Word Benchmark (Chelba et al., 2014) is 39.7, compared to 30.0 for the forward `CNN-BIG-LSTM`. Generally, we found the forward and backward perplexities to be approximately equal, with the backward value slightly lower.

Fine tuning on task specific data resulted in significant drops in perplexity and an increase in downstream task performance in some cases. This can be seen as a type of domain transfer for the biLM. As a result, in most cases we used a fine-tuned biLM in the downstream task. See the Appendix for details.

## 4 Evaluation

Table 1 shows the performance of ELMo across a diverse set of six benchmark NLP tasks. In every task considered, simply adding ELMo establishes a new state-of-the-art result, with relative error reductions ranging from 6 - 20% over strong base models. This is a very general result across a diverse set model architectures and language understanding tasks. In the remainder of this section we provide high-level sketches of the individual task results; see the Appendix for full experimental details.

Table 1: Test set comparison of ELMo enhanced neural models with state-of-the-art single model baselines across six benchmark NLP tasks. The performance metric varies across tasks – accuracy for SNLI and SST-5; $F_1$ for SQuAD, SRL and NER; average $F_1$ for Coref. Due to the small test sizes for NER and SST-5, we report the mean and standard deviation across five runs with different random seeds. The "increase" column lists both the absolute and relative improvements over our baseline.

| TASK | PREVIOUS SOTA | | OUR BASELINE | ELMO + BASELINE | INCREASE (ABSOLUTE/ RELATIVE) |
|---|---|---|---|---|---|
| SNLI | McCann et al. (2017) | 88.1 | 88.0 | $88.7 \pm 0.17$ | 0.7 / 5.8% |
| SQuAD[2] | r-net Wang et al. (2017) | 84.3 | 81.1 | 85.3 | 4.2 / 22.2% |
| SRL | He et al. (2017) | 81.7 | 81.4 | 84.6 | 3.2 / 17.2% |
| Coref | Lee et al. (2017) | 67.2 | 67.2 | 70.4 | 3.2 / 9.8% |
| NER | Peters et al. (2017) | $91.93 \pm 0.19$ | 90.15 | $92.22 \pm 0.10$ | 2.06 / 21% |
| SST-5 | McCann et al. (2017) | 53.7 | 51.4 | $54.7 \pm 0.5$ | 3.3 / 6.8% |

**Textual entailment** Textual entailment is the task of determining whether a "hypothesis" is true, given a "premise". The Stanford Natural Language Inference (SNLI) corpus (Bowman et al., 2015) provides approximately 550K hypothesis/premise pairs. Our baseline, the ESIM sequence model from Chen et al. (2017), uses a biLSTM to encode the premise and hypothesis, followed by a matrix attention layer, a local inference layer, another biLSTM inference composition layer, and finally a pooling operation before the output layer. Overall, adding ELMo to the ESIM model improves accuracy by an average of 0.7% across five random seeds, increasing the single model state-of-the-art by 0.6% over the CoVe enhanced model from McCann et al. (2017). A five member ensemble pushes the overall accuracy to 89.3%, exceeding the previous ensemble best of 88.9% (Gong et al., 2017) – see Appendix for details.

**Question answering** The Stanford Question Answering Dataset (SQuAD) (Rajpurkar et al., 2016) contains 100K+ crowd sourced question-answer pairs where the answer is a span in a given Wikipedia paragraph. Our baseline model (Clark & Gardner, 2017) is an improved version of the Bidirectional Attention Flow model in Seo et al. (BiDAF; 2017). It adds a self-attention layer after the bidirectional attention component, simplifies some of the pooling operations and substitutes the LSTMs for gated recurrent units (GRUs; Cho et al., 2014). After adding ELMo to the baseline model, test set $F_1$ improved by 4.2% from 81.1% to 85.3%, improving the single model state-of-the-art by 1.0%.

**Semantic role labeling** A semantic role labeling (SRL) system models the predicate-argument structure of a sentence, and is often described as answering "Who did what to whom". SRL is a challenging NLP task as it requires jointly extracting the arguments of a predicate and establishing their semantic roles. He et al. (2017) modeled SRL as a BIO tagging problem and used an 8-layer deep biLSTM with forward and backward directions interleaved, following Zhou & Xu (2015). As shown in Table 1, when adding ELMo to a re-implementation of He et al. (2017) the single model test set $F_1$ jumped 3.2% from 81.4% to 84.6% – a new state-of-the-art on the OntoNotes benchmark (Pradhan et al., 2013), even improving over the previous best ensemble result by 1.2% (see Table 10 in the Appendix).

**Coreference resolution** Coreference resolution is the task of clustering mentions in text that refer to the same underlying real world entities. Our baseline model is the end-to-end span-based neural model of Lee et al. (2017). It uses a biLSTM and attention mechanism to first compute span representations and then applies a softmax mention ranking model to find coreference chains. In our experiments with the OntoNotes coreference annotations from the CoNLL 2012 shared task (Pradhan et al., 2012), adding ELMo improved the average $F_1$ by 3.2% from 67.2 to 70.4, establishing a new state of the art, again improving over the previous best ensemble result by 1.6% $F_1$ (see Table 11 in the Appendix).

**Named entity extraction** The CoNLL 2003 NER task (Sang & Meulder, 2003) consists of newswire from the Reuters RCV1 corpus tagged with four different entity types (`PER`, `LOC`, `ORG`, `MISC`). Fol-

---

[2]As of October 22, 2017 (`https://rajpurkar.github.io/SQuAD-explorer/`)

Table 2: Development set performance for SQuAD, SNLI and SRL comparing using all layers of the biLM (with different choices of regularization strength $\lambda$) to just the top layer.

| Task | Baseline | Last Only | All layers | |
|------|----------|-----------|-----------|-----------|
| | | | $\lambda$=1 | $\lambda$=0.001 |
| SQuAD | 80.8 | 82.5 | 83.6 | **84.8** |
| SNLI | 88.1 | 89.1 | 89.3 | **89.5** |
| SRL | 81.6 | 84.1 | 84.6 | **84.8** |

Table 3: Development set performance for SQuAD, SNLI and SRL when including ELMo at different locations in the supervised model.

| Task | Input Only | Input & Output | Output Only |
|------|-----------|----------------|-------------|
| SQuAD | 84.2 | **84.8** | 83.7 |
| SNLI | 88.9 | **89.5** | 88.7 |
| SRL | **84.7** | 84.3 | 80.9 |

lowing recent state-of-the-art systems (Lample et al., 2016; Peters et al., 2017), the baseline model is a biLSTM-CRF based sequence tagger. It forms a token representation by concatenating pre-trained word embeddings with a character-based CNN representation, passes it through two layers of biL-STMs, and then computes the sentence conditional random field (CRF) loss (Lafferty et al., 2001) during training and decodes with the Viterbi algorithm during testing, similar to Collobert et al. (2011). As shown in Table 1, our ELMo enhanced biLSTM-CRF achieves 92.22% $F_1$ averaged over five runs. The key difference between our system and the previous state of the art from Peters et al. (2017) is that we allowed the task model to learn a weighted average of all biLM layers, whereas Peters et al. (2017) only use the top biLM layer. As shown in Sec. 5.1, using all layers instead of just the last layer improves performance across multiple tasks.

**Sentiment analysis** The fine-grained sentiment classification task in the Stanford Sentiment Treebank (SST-5; Socher et al., 2013) involves selecting one of five labels (from very negative to very positive) to describe a sentence from a movie review. The sentences contain diverse linguistic phenomena such as idioms, named entities related to film, and complex syntactic constructions (e.g., negations) that are difficult for models to learn directly from the training dataset alone. Our baseline model is the biattentive classification network (BCN) from McCann et al. (2017), which also held the prior state-of-the-art result when augmented with CoVe embeddings. Replacing CoVe with ELMo in the BCN model results in a 1.0% absolute accuracy improvement over the state of the art.

## 5 ANALYSIS

This section provides an ablation analysis to validate our chief claims and to elucidate some interesting aspects of ELMo representations. Sec. 5.1 shows that using deep contextual representations in downstream tasks improves performance over previous work that uses just the top layer, regardless of whether they are produced from a biLM or MT encoder, and that ELMo representations provide the best overall performance. Sec. 5.3 explores the different types of contextual information captured in biLMs and confirms that syntactic information is better represented at lower layers while semantic information is captured a higher layers, consistent with MT encoders. It also shows that our biLM consistently provides richer representations then CoVe. Additionally, we analyze the sensitivity to where ELMo is included in the task model (Sec. 5.2), training set size (Sec. 5.4), and visualize the ELMo learned weights across the tasks (Sec. 5.5).

### 5.1 ALTERNATE LAYER WEIGHTING SCHEMES

There are many alternatives to Equation 1 for combining the biLM layers. Previous work on contextual representations use only the last layer, whether it be from a biLM (Peters et al., 2017) or an MT encoder (CoVe; McCann et al., 2017). The choice of the regularization parameter $\lambda$ is also important, as large values such as $\lambda = 1$ effectively reduce the weighting function to a simple average over the layers, while smaller values (e.g., $\lambda = 0.001$) allows the layer weights to vary.

Table 2 compares these alternatives for SNLI, SRL and SQuAD. Including representations from all layers improves overall performance over just using the last layer, and including contextual representations from the last layer improves performance over the baseline. For example, in the case of SQuAD, using just the last biLM layer improves development $F_1$ by 1.7% over the baseline. Aver-

Table 4: Nearest neighbors to "play" using GloVe and the context embeddings from a biLM.

| | Source | Nearest Neighbors |
|---|---|---|
| GloVe | play | playing, game, games, played, players, plays, player, Play, football, multiplayer |
| biLM | Chico Ruiz made a spectacular play on Alusik 's grounder {...} | Kieffer , the only junior in the group , was commended for his ability to hit in the clutch , as well as his all-round excellent play . |
| | Olivia De Havilland signed to do a Broadway play for Garson {...} | {...} they were actors who had been handed fat roles in a successful play , and had talent enough to fill the roles competently , with nice understatement . |

aging all biLM layers instead of using just the last layer improves $F_1$ another 1.1% (comparing "Last Only" to $\lambda=1$ columns), and allowing the task model to learn individual layer weights improves $F_1$ another 1.2% ($\lambda=1$ vs. $\lambda=0.001$). A small $\lambda$ is preferred in most cases with ELMo, although for NER, a task with a smaller training set, the results are insensitive to $\lambda$ (not shown).

The overall trend is similar with CoVe but with smaller increases over the baseline. In the case of SNLI, weighting all layers with $\lambda = 1$ improves development accuracy from 88.2 to 88.7% over using just the last layer. SRL $F_1$ increased a marginal 0.1% to 82.2 for the $\lambda = 1$ case compared to using the last layer only.

## 5.2 WHERE TO INCLUDE ELMO?

All of the task architectures in this paper include word embeddings only as input to the lowest layer biRNN. However, we find that including ELMo at the output of the biRNN in task-specific architectures improves overall results for some tasks. As shown in Table 3, including ELMo at both the input and output layers for SNLI and SQuAD improves over just the input layer, but for SRL (and coreference resolution, not shown) performance is highest when it is included at just the input layer. One possible explanation for this result is that both the SNLI and SQuAD architectures use attention layers after the biRNN, so introducing ELMo at this layer allows the supervised model to attend directly to the biLM's internal representations. In the SRL case, the task-specific context representations are likely more important than those from the biLM.

## 5.3 WHAT INFORMATION IS CAPTURED BY THE BILM'S REPRESENTATIONS?

Since adding ELMo improves task performance over word vectors alone, the biLM's contextual representations must encode information generally useful for NLP tasks that is not captured in word vectors. Intuitively, the biLM must be disambiguating the meaning of words using their context. Consider "play", a highly polysemous word. The top of Table 4 lists nearest neighbors to "play" using GloVe vectors. They are spread across several parts of speech (e.g., "played", "playing" as verbs, and "player", "game" as nouns) but concentrated in the sports-related senses of "play". In contrast, the bottom two rows show nearest neighbor sentences from the SemCor dataset (see below) using the biLM's context representation of "play" in the source sentence. In these cases, the biLM is able to disambiguate both the part of speech and word sense in the source sentence.

These observations can be quantified using an approach similar to Belinkov et al. (2017). To isolate the information encoded by the biLM, the representations are used to directly make predictions for a fine grained word sense disambiguation (WSD) task and a POS tagging task. Using this approach, it is also possible to compare to CoVe, and across each of the individual layers.

**Word sense disambiguation** Given a sentence, we can use the biLM representations to predict the sense of a target word using a simple 1-nearest neighbor approach, similar to Melamud et al. (2016). To do so, we first use the biLM to compute representations for all words in SemCor 3.0, our training corpus (Miller et al., 1994), and then take the average representation for each sense. At test time, we again use the biLM to compute representations for a given target word and take the nearest neighbor sense from the training set, falling back to the first sense from WordNet for lemmas not observed during training.

Table 5: All-words fine grained WSD $F_1$. For CoVe and the biLM, we report scores for both the first and second layer biLSTMs.

| Model | $F_1$ |
|---|---|
| WordNet 1st Sense Baseline | 65.9 |
| Raganato et al. (2017a) | 69.9 |
| Iacobacci et al. (2016) | **70.1** |
| CoVe, First Layer | 59.4 |
| CoVe, Second Layer | *64.7* |
| biLM, First layer | 67.4 |
| biLM, Second layer | *69.0* |

Table 6: Test set POS tagging accuracies for PTB. For CoVe and the biLM, we report scores for both the first and second layer biLSTMs.

| Model | Acc. |
|---|---|
| Collobert et al. (2011) | 97.27 |
| Ma & Hovy (2016) | 97.55 |
| Ling et al. (2015) | **97.78** |
| CoVe, First Layer | *93.3* |
| CoVe, Second Layer | 92.8 |
| biLM, First Layer | *97.0* |
| biLM, Second Layer | 95.8 |

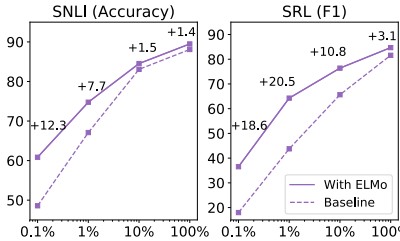

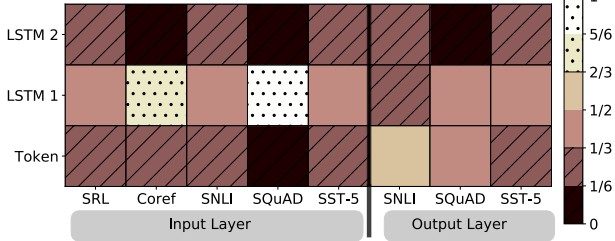

Figure 1: Comparison of baseline vs. ELMo performance for SNLI and SRL as the training set size is varied from 0.1% to 100%.

Figure 2: Visualization of softmax normalized biLM layer weights across tasks and ELMo locations. Normalized weights less then $1/3$ are hatched with horizontal lines and those greater then $2/3$ are speckled.

Table 5 compares WSD results using the evaluation framework from Raganato et al. (2017b) across the same suite of four test sets in Raganato et al. (2017a). Overall, the biLM top layer representations have $F_1$ of 69.0 and are better at WSD then the first layer. This is competitive with a state-of-the-art WSD-specific supervised model using hand crafted features (Iacobacci et al., 2016) and a task specific biLSTM that is also trained with auxiliary coarse-grained semantic labels and POS tags (Raganato et al., 2017a). The CoVe biLSTM layers follow a similar pattern to those from the biLM (higher overall performance at the second layer compared to the first); however, our biLM outperforms the CoVe biLSTM, which trails the WordNet first sense baseline.

**POS tagging** To examine whether the biLM captures basic syntax, we used the context representations as input to a linear classifier that predicts POS tags with the Wall Street Journal portion of the Penn Treebank (PTB) (Marcus et al., 1993). As the linear classifier adds only a tiny amount of model capacity, this is direct test of the biLM's representations. Similar to WSD, the biLM representations are competitive with carefully tuned, task specific biLSTMs with character representations (Ling et al., 2015; Ma & Hovy, 2016). However, unlike WSD, accuracies using the first biLM layer are higher than the top layer, consistent with results from deep biLSTMs in multi-task training (Søgaard & Goldberg, 2016; Hashimoto et al., 2017) and MT (Belinkov et al., 2017). CoVe POS tagging accuracies follow the same pattern as those from the biLM, and just like for WSD, the biLM achieves higher accuracies than the CoVe encoder.

**Implications for supervised tasks** Taken together, these experiments confirm different layers in the biLM represent different types of information and explain why including all biLM layers is important for the highest performance in downstream tasks. In addition, the biLM's representations are more transferable to WSD and POS tagging than those in CoVe, which helps illustrate why ELMo outperforms CoVe in downstream tasks.

## 5.4 SAMPLE EFFICIENCY

Adding ELMo to a model increases the sample efficiency considerably, both in terms of number of parameter updates to reach state-of-the-art performance and the overall training set size. For

example, the SRL model reaches a maximum development $F_1$ after 486 epochs of training without ELMo. After adding ELMo, the model exceeds the baseline maximum at epoch 10, a 98% relative decrease in the number of updates needed to reach the same level of performance.

In addition, ELMo-enhanced models use smaller training sets more efficiently than models without ELMo. Figure 1 compares the performance of baselines models with and without ELMo as the percentage of the full training set is varied from 0.1% to 100%. Improvements with ELMo are largest for smaller training sets and significantly reduce the amount of training data needed to reach a given level of performance. In the SRL case, the ELMo model with 1% of the training set has about the same $F_1$ as the baseline model with 10% of the training set.

### 5.5 VISUALIZATION OF LEARNED WEIGHTS

Figure 2 visualizes the softmax-normalized learned layer weights across the tasks. At the input layer, in all cases, the task model favors the first biLSTM layer, with the remaining emphasis split between the token layer and top biLSTM in task specific ways. For coreference and SQuAD, the first LSTM layer is strongly favored, but the distribution is less peaked for the other tasks. It is an interesting question for future work to understand why the first biLSTM layer is universally favored. The output layer weights are relatively balanced, with a slight preference for the lower layers.

## 6 CONCLUSION AND FUTURE WORK

We have introduced a general approach for learning high-quality deep context-dependent representations from biLMs, and shown large improvements when applying ELMo to a broad range of NLP tasks. Through ablations and other controlled experiments, we have also confirmed that the biLM layers efficiently encode different types of syntactic and semantic information about words-in-context, and that using all layers improves overall task performance.

Our approach raises several interesting questions for future work, broadly organized into two themes.

**"What is the best training regime for learning generally useful NLP representations?"** By choosing a biLM training objective, we benefit from nearly limitless unlabeled text and can immediately apply advances in language modeling, an active area of current research. However, it's possible that further decreases in LM perplexity will not translate to more transferable representations, and that other objective functions might be more suitable for learning general purpose representations.

**"What is the best way to use deep contextual representations for other tasks?"** Our method of using a weighted average of all layers from the biLM is simple and empirically successful. However, a deeper fusion of the biLM layers with a target NLP architecture may lead to further improvements.

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

# 7 APPENDIX

This Appendix contains details of the model architectures, training routines and hyper-parameter choices for the state-of-the-art models in Section 4.

All of the individual models share a common architecture in the lowest layers with a context independent token representation below several layers of stacked RNNs – LSTMs in every case except the SQuAD model that uses GRUs.

## 7.1 FINE TUNING BILM

As noted in Sec. 3.4, fine tuning the biLM on task specific data typically resulted in significant drops in perplexity. To fine tune on a given task, the supervised labels were temporarily ignored, the biLM fine tuned for one epoch on the training split and evaluated on the development split. Once fine tuned, the biLM weights were fixed during task training.

Table 7 lists the development set perplexities for the considered tasks. In every case except CoNLL 2012, fine tuning results in a large improvement in perplexity, e.g., from 72.1 to 16.8 for SNLI.

The impact of fine tuning on supervised performance is task dependent. In the case of SNLI, fine tuning the biLM increased development accuracy 0.6% from 88.9% to 89.5% for our single best model. However, for sentiment classification development set accuracy is approximately the same regardless whether a fine tuned biLM was used.

## 7.2 IMPORTANCE OF $\gamma$ IN EQN. (1)

The $\gamma$ parameter in Eqn. (1) was of practical importance to aid optimization, due to the different distributions between the biLM internal representations and the task specific representations. It is especially important in the last-only case in Sec. 5.1. Without this parameter, the last-only case performed poorly (well below the baseline) for SNLI and training failed completely for SRL.

## 7.3 TEXTUAL ENTAILMENT

Our baseline SNLI model is the ESIM sequence model from Chen et al. (2017). Following the original implementation, we used 300 dimensions for all LSTM and feed forward layers and pre-trained 300 dimensional GloVe embeddings that were fixed during training. For regularization, we added 50% variational dropout (Gal & Ghahramani, 2016) to the input of each LSTM layer and 50% dropout (Srivastava et al., 2014) at the input to the final two fully connected layers. All feed forward layers use ReLU activations. Parameters were optimized using Adam (Kingma & Ba, 2015) with gradient norms clipped at 5.0 and initial learning rate 0.0004, decreasing by half each time accuracy on the development set did not increase in subsequent epochs. The batch size was 32.

The best ELMo configuration added ELMo vectors to both the input and output of the lowest layer LSTM, using (1) with layer normalization and $\lambda = 0.001$. Due to the increased number of parameters in the ELMo model, we added $\ell^2$ regularization with regularization coefficient 0.0001 to all recurrent and feed forward weight matrices and 50% dropout after the attention layer.

Table 8 compares test set accuracy of our system to previously published systems. Overall, adding ELMo to the ESIM model improved accuracy by 0.7% establishing a new single model state-of-the-art of 88.7%, and a five member ensemble pushes the overall accuracy to 89.3%.

## 7.4 QUESTION ANSWERING

Our QA model is a simplified version of the model from Clark & Gardner (2017). It embeds tokens by concatenating each token's case-sensitive 300 dimensional GloVe word vector (Pennington et al., 2014) with a character-derived embedding produced using a convolutional neural network followed by max-pooling on learned character embeddings. The token embeddings are passed through a shared bi-directional GRU, and then the bi-directional attention mechanism from BiDAF Seo et al. (2017). The augmented context vectors are then passed through a linear layer with ReLU activations, a residual self-attention layer that uses a GRU followed by the same attention mechanism

Table 7: Development set perplexity before and after fine tuning for one epoch on the training set for various datasets (lower is better). Reported values are the average of the forward and backward perplexities.

| Dataset | | Before tuning | After tuning |
|---|---|---|---|
| SNLI | | 72.1 | 16.8 |
| CoNLL 2012 (coref/SRL) | | 92.3 | - |
| CoNLL 2003 (NER) | | 103.2 | 46.3 |
| SQuAD | Context | 99.1 | 43.5 |
| | Questions | 158.2 | 52.0 |
| SST | | 131.5 | 78.6 |

applied context-to-context, and another linear layer with ReLU activations. Finally, the results are fed through linear layers to predict the start and end token of the answer.

Variational dropout is used before the input to the GRUs and the linear layers at a rate of 0.2. A dimensionality of 90 is used for the GRUs, and 180 for the linear layers. We optimize the model using Adadelta with a batch size of 45. At test time we use an exponential moving average of the weights and limit the output span to be of at most size 17. We do not update the word vectors during training.

Performance was highest when adding ELMo without layer normalization to both the input and output of the contextual GRU layer and leaving the ELMo weights unregularized ($\lambda = 0$).

Table 9 compares single model test set results as of October 22, 2017, taken from the SQuAD leaderboard. Overall, our submission was the highest single model result, improving the previous single model result (r-net) by 1.0% $F_1$ and our baseline by 4.2%. Note the model from Clark & Gardner (2017) additionally uses GRUs for prediction, and is stronger than our model without ELMo (79.4 vs 80.7 F1 on the dev set), so we anticipate being able to make further performance improvements by incorporating these changes into our design.

## 7.5 SEMANTIC ROLE LABELING

Our baseline SRL model is an exact reimplementation of (He et al., 2017). Words are represented using a concatenation of 100 dimensional vector representations, initialized using GloVe (Pennington et al., 2014) and a binary, per-word predicate feature, represented using an 100 dimensional embedding. This 200 dimensional token representation is then passed through an 8 layer "interleaved" biLSTM with a 300 dimensional hidden size, in which the directions of the LSTM layers alternate per layer. This deep LSTM uses Highway connections (Srivastava et al., 2015) between layers and variational recurrent dropout (Gal & Ghahramani, 2016). This deep representation is then projected using a final dense layer followed by a softmax activation to form a distribution over all possible tags. Labels consist of semantic roles from PropBank (Palmer et al., 2005) augmented with a BIO labeling scheme to represent argument spans. During training, we minimize the negative log likelihood of the tag sequence using Adadelta with a learning rate of 1.0 and $\rho = 0.95$ (Zeiler, 2012). At test time, we perform Viterbi decoding to enforce valid spans using BIO constraints. Variational dropout of 10% is added to all LSTM hidden layers. Gradients are clipped if their value exceeds 1.0. Models are trained for 500 epochs or until validation F1 does not improve for 200 epochs, whichever is sooner. The pretrained GloVe vectors are fine-tuned during training. The final dense layer and all cells of all LSTMs are initialized to be orthogonal. The forget gate bias is initialized to 1 for all LSTMs, with all other gates initialized to 0, as per (Józefowicz et al., 2015).

Table 10 compares test set F1 scores of our ELMo augmented implementation of (He et al., 2017) with previous results. Our single model score of 84.6 F1 represents a new state-of-the-art result on the CONLL 2012 Semantic Role Labeling task, surpassing the previous single model result by 2.9 F1 and a 5-model ensemble by 1.2 F1.

Table 8: SNLI test set accuracy.[4]Single model results occupy the portion, with ensemble results at the bottom.

| Model | Acc. |
|---|---|
| Feature based (Bowman et al., 2015) | 78.2 |
| DIIN (Gong et al., 2017) | 88.0 |
| BCN+Char+CoVe (McCann et al., 2017) | 88.1 |
| ESIM (Chen et al., 2017) | 88.0 |
| ESIM+ELMo | **88.7** $\pm$ 0.17 |
| ESIM+TreeLSTM (Chen et al., 2017) | 88.6 |
| DIIN ensemble (Gong et al., 2017) | 88.9 |
| ESIM+ELMo ensemble | **89.3** |

### 7.6 COREFERENCE RESOLUTION

Our baseline coreference model is the end-to-end neural model from Lee et al. (2017) with all hyperparameters exactly following the original implementation.

The best configuration added ELMo to the input of the lowest layer biLSTM and weighted the biLM layers using (1) without any regularization ($\lambda = 0$) or layer normalization. 50% dropout was added to the ELMo representations.

Table 11 compares our results with previously published results. Overall, we improve the single model state-of-the-art by 3.2% average $F_1$, and our single model result improves the previous ensemble best by 1.6% $F_1$. Adding ELMo to the output from the biLSTM in addition to the biLSTM input reduced $F_1$ by approximately 0.7% (not shown).

### 7.7 NAMED ENTITY RECOGNITION

Our baseline NER model concatenates 50 dimensional pre-trained Senna vectors (Collobert et al., 2011) with a CNN character based representation. The character representation uses 16 dimensional character embeddings and 128 convolutional filters of width three characters, a ReLU activation and by max pooling. The token representation is passed through two biLSTM layers, the first with 200 hidden units and the second with 100 hidden units before a final dense layer and softmax layer. During training, we use a CRF loss and at test time perform decoding using the Viterbi algorithm while ensuring that the output tag sequence is valid.

Variational dropout is added to the input of both biLSTM layers. During training the gradients are rescaled if their $\ell^2$ norm exceeds 5.0 and parameters updated using Adam with constant learning rate of 0.001. The pre-trained Senna embeddings are fine tuned during training. We employ early stopping on the development set and report the averaged test set score across five runs with different random seeds.

ELMo was added to the input of the lowest layer task biLSTM. As the CoNLL 2003 NER data set is relatively small, we found the best performance by constraining the trainable layer weights to be effectively constant by setting $\lambda = 0.1$ with (1).

Table 12 compares test set $F_1$ scores of our ELMo enhanced biLSTM-CRF tagger with previous results. Overall, the 92.22% $F_1$ from our system establishes a new state-of-the-art. When compared to Peters et al. (2017), using representations from all layers of the biLM provides a modest improvement.

### 7.8 SENTIMENT CLASSIFICATION

We use almost the same biattention classification network architecture described in McCann et al. (2017), with the exception of replacing the final maxout network with a simpler feedforward network composed of two ReLu layers with dropout. A BCN model with a batch-normalized maxout network reached significantly lower validation accuracies in our experiments, although there may

---

[4]A comprehensive comparison can be found at `https://nlp.stanford.edu/projects/snli/`

Table 9: Single model test set results for SQuAD, showing both Exact Match (EM) and $F_1$. References provided where available.

| Model | EM | $F_1$ |
|---|---|---|
| BiDAF (Seo et al., 2017) | 68.0 | 77.3 |
| BiDAF + Self Attention | 72.1 | 81.1 |
| DCN+ | 75.1 | 83.1 |
| Reg-RaSoR | 75.8 | 83.3 |
| AIR-FusionNet | 76.0 | 83.9 |
| r-net Wang et al. (2017) | 76.5 | 84.3 |
| BiDAF + Self Attention + ELMo | **77.9** | **85.3** |

Table 10: SRL CoNLL 2012 test set $F_1$.

| Model | $F_1$ |
|---|---|
| Pradhan et al. (2013) | 77.5 |
| Zhou & Xu (2015) | 81.3 |
| (He et al., 2017), single | 81.7 |
| (He et al., 2017), ensemble | 83.4 |
| (He et al., 2017), our impl. | 81.4 |
| (He et al., 2017) + ELMo | **84.6** |

Table 11: Coreference resolution average $F_1$ on the test set from the CoNLL 2012 shared task.

| Model | Average $F_1$ |
|---|---|
| Durrett & Klein (2013) | 60.3 |
| Wiseman et al. (2016) | 64.2 |
| (Clark & Manning, 2016) | 65.7 |
| (Lee et al., 2017) (single) | 67.2 |
| (Lee et al., 2017) (ensemble) | 68.8 |
| (Lee et al., 2017) + ELMo | **70.4** |

Table 12: Test set $F_1$ for CoNLL 2003 NER task. Models with ♣ included gazetteers and those with ◇ used both the train and development splits for training.

| Model | $F_1 \pm$ std. |
|---|---|
| Collobert et al. (2011)♣ | 89.59 |
| Lample et al. (2016) | 90.94 |
| Ma & Hovy (2016) | 91.2 |
| (Chiu & Nichols, 2016)♣,◇ | $91.62 \pm 0.33$ |
| (Peters et al., 2017)◇ | $91.93 \pm 0.19$ |
| biLSTM-CRF + ELMo | $\mathbf{92.22 \pm 0.10}$ |

Table 13: Test set accuracy for SST-5.

| Model | Acc. |
|---|---|
| DMN (Kumar et al., 2016) | 52.1 |
| LSTM-CNN (Zhou et al., 2016) | 52.4 |
| NTI (Munkhdalai & Yu, 2017) | 53.1 |
| BCN+Char+CoVe (McCann et al., 2017) | 53.7 |
| BCN+ELMo | **54.7** |

be discrepancies between our implementation and that of McCann et al. (2017). To match the CoVe training setup, we only train on phrases that contain four or more tokens. We use 300-d hidden states for the biLSTM and optimize the model parameters with Adam Kingma & Ba (2015) using a learning rate of 0.0001. The trainable biLM layer weights are regularized by $\lambda = 0.001$, and we add ELMo to both the input and output of the biLSTM; the output ELMo vectors are computed with a second biLSTM and concatenated to the input.

