# OpenReview forum: "Deep contextualized word representations"
_ICLR.cc/2018/Conference — Accept (Poster)_

### Official Review · AnonReviewer1 · 2017-11-27
**Learning contextual embeddings for tokens can improve accuracies in language tasks**

**Rating:** 6
**Confidence:** 4

**Review:**

The authors learn token embeddings that use surrounding context by concatenating representations obtained by training a bidirectional language model, very similar to Peters et al. 2017. They learn a distribution of weights for each layer of embeddings of the pre-trained bi-lm language model. These embeddings improve accuracies over a large range of tasks.

The technical contribution of the paper seems minimal on top of Peters et al.: Learning weights for every layer of embeddings, adding dropout, and adding a regularization term to the training. They have evaluated the efficacy of these embeddings on more tasks than the original paper that introduced them. As it stands, there is not enough novelty in this paper. Answering the questions from their future work would be interesting and would add more technical depth to the paper.

---

> ### Author Response · Authors · 2017-12-07
> **Response**
>
> We disagree about the lack of novelty from Peters et al. (2017).  The BiLM architecture is unified and improved, as compared to Peters et al., who used off-the-shelf independently trained left-to-right and right-to-left models. Furthermore, exposing the representations from all layers in the biLM to the task models via our scalar weighting function is a completely new idea. We show it is crucial to obtain the best performance, as demonstrated in Table 2.  Overall, our method works extremely well in practice, despite its technical simplicity. We think this fact should be considered a major strength of the work.

---

### Official Review · AnonReviewer3 · 2017-11-27
**Promising results**

**Rating:** 6
**Confidence:** 3

**Review:**

This paper proposed a model called ‘deep contextualized’ to extract word embeddings for downstream applications. This model is simply a bi-directional language model (biLM) and the word embedding is a weighted combination of the output of the hidden layers (forward and backward). Furthermore like previous work, the authors proposed to pre-train the biLM with a large amount of data and then use the embeddings in combination with the context-independent word embedding in neural network models (in this paper, RNN) for final applications. Their results showed consistent improvements over the baseline and the previous best systems on several tasks.

Some detailed comments:
-	I would like to see an overview figure which illustrates the biLM and its integration in downstream applications. Furthermore, it is also interesting to see the performance difference between with and without fine-tuning of the biLM.
-	Intrinsic evaluations are missing in this paper. Although we know that word similarity tasks are not the best way to evaluate word embeddings, it is always informative to report results on these standard tasks.
-	The baseline results in table 1 and table 2 are not consistent. Is there something wrong?
-	Did the baseline systems use pretrained embeddings or randomly initialized ones? In my opinion, the right baseline systems should use pretrained word2vec or glove embeddings.

---

> ### Author Response · Authors · 2017-12-07
> **Response**
>
> Our overall goal in this work is to improve performance on a wide range of NLP tasks, hence the primary focus on extrinsic evaluation.  However, we did provide intrinsic evaluations of the biLM’s contextual representations compared to those from a NMT system in Section 5.3.  As contextual representations are fundamentally different than traditional word embeddings, they require a different set of intrinsic evaluations.  Moreover, it’s likely that the WSD and POS evaluations in Section 5.3 are better correlated with extrinsic performance than the traditional intrinsic word embedding evaluations. This is an interesting area for future work.
>
> Minor points:
> 1.  “The baseline results in table 1 and table 2 are not consistent. Is there something wrong?”  As mentioned in the captions, Table 1 evaluates on the test set, while Table 2 shows ablations on the development set.
> 2.  “Did the baseline systems use pretrained embeddings or randomly initialized ones? In my opinion, the right baseline systems should use pretrained word2vec or glove embeddings.”  The baseline systems are state-of-the-art systems.  They all use pre-trained embeddings as described in Section 3.3 and the model details in the Appendix.

---

### Official Review · AnonReviewer2 · 2017-11-28

**Rating:** 5
**Confidence:** 5

**Review:**

This paper proposes a method to learn contextualized word representations (ELMO) by pretraining a multilayer bidirectional LSTM language model and using representations from all levels of the LSTM in the input or output layer of a supervised task of interest.
Experiments on various datasets (SNLI, SQuAD, SRL, Coref, NER, SST) show that the proposed method improve over baseline models.
Ablation analysis demonstrate that using all layers of ELMO is always better than just using only the final layer, and that representations learned by ELMO capture basic notions of word senses and part of speeches.

The paper is well written and I think learning contextualized word representations is an important topic.
However, one thing that I am not sure about from experiments in the paper is whether the improvements come from an increase in model capacity and (unlabeled) data used to train the model, or whether there are more interesting things going on.
- What makes the proposed approach different than just a deeper architecture for each of the considered tasks, where some parts of the network are trained using unlabeled data?
- Is the pretraining with unlabeled data necessary, or can we just have this deep architecture and train everything with the available supervised data?
- An ELMO enhanced model has more parameters than the baseline model for each task. What is the performance of the (non deep) baseline method with comparable number of parameters (bigger hidden size)?

More generally, it is not surprising that given sufficient training data, a deeper model (e.g., ELMO enhanced models) with multiple connections across layers will perform better than shallower models with fewer parameters.
I would like to see more analysis and/or explanations on why the proposed method contributes more beyond this.

---

> ### Author Response · Authors · 2017-12-07
> **Response**
>
> The improvements in our paper are not just due to increasing model capacity.  The contextual ELMo representations are a fundamentally different type of word representation then GloVe and word2vec.  The biLM encodes information not available by simply increasing the hidden size of current supervised models and allowing supervised models to preferentially access the internal representations provides the best performance.
>
> We can conclude this because:
> 1.  Our baseline models are state-of-the-art systems that have been carefully tuned by their respective authors.  If solely increasing model capacity improved performance the original authors would have done so.
> 2.  Adding ELMo improved performance significantly on every considered NLP task, generally with 10-20% relative error reduction.
> 3.  In our ablation results (Table 2), the combination of all biLM layers is essential for the highest performance.
> 4.  Pre-training with unlabeled data is necessary, as shown in previous work (Peters et al 2017 showed that training the language model at a large scale on a large unsupervised corpora was crucial for the NER task).
>
> Certainly, the unsupervised biLM objective allows us to train a high capacity model, similar to traditional word embeddings.  However, we note the pretrained component of the biLM used to compute ELMo representations has about 93 million parameters, whereas the 300 dimensional GloVe vectors have about 650 million parameters.  ELMo’s advantage is in the contextual nature of the representations and the manner in which we integrated it in the downstream task, not from increased capacity over GloVe.
>
> Conceptually, our proposed approach is similar to a deep multitask architecture across tasks, where a piece of the architecture is shared and trained on unlabeled data (the ELMo piece).  However, that still leaves a huge space of possible architectures.  Concretely, one of our paper’s contributions is to propose a particular instantiation of this architecture that works extremely well in practice.

---

### Public Comment · ~Samuel_R._Bowman1 · 2017-11-30
**Neat results! Minor comment.**

Your claim about the SotA on SNLI isn't quite right. What to use depends on what you think about ensembles, but there's a leaderboard up here: https://nlp.stanford.edu/projects/snli/

The ESIM paper from 2016 reports slightly better numbers than the 2017 McCann paper you cite as SotA.

Overall, though, these are exciting results, and I'd be interested in playing with the trained model(s) once they come out.

---

> ### Author Response · Authors · 2017-11-30
> **Previous SNLI SotA**
>
> Glad you enjoyed the paper!  We are working on the code and should have an initial version to compute ELMo representations out in the next week or two.
>
> Thanks for your comment about the previous SNLI single model SotA.  We'll change Table 1 to replace the reference to McCann 2017 with the ESIM + 300D Syntactic TreeLSTM (88.6) from Chen et al. 2017.  In our draft we had classified this model as an ensemble method (see Table 8 in Appendix) but will update in the next version.

---

> > ### Public Comment · ~Samuel_R._Bowman1 · 2017-11-30
> > **Quick reply**
> >
> > That's entirely reasonable—it is somewhat ensemble-like.

---

### Public Comment · (anonymous) · 2017-12-04
**Question from a novice**

Hi, I'm a student conducting a deep learning project on NMT and I'm trying to implement your ELMo enhenced embeddings.

I'm a bit confused about the dimension of the input. If I'm to concat x_k(1x512) with ELMo_k(1x512) to be used as new input, does it mean that I have to modify my model to take 1X1024 sized input?

And I haven't clearly understood about appeding ELMo_k on the output side of the task model.
1. By task model do you mean the biLM?
2. Or the model that actually gets the task done? (in my case NMT)
In either case, could you elaborate more on "including ELMo at the output of task RNN"?

Thank you for the great paper.

---

> ### Author Response · Authors · 2017-12-07
> **Implementation details**
>
> Thanks for your interest in our work!  To answer your questions:
>
> "If I'm to concat x_k(1x512) with ELMo_k(1x512) to be used as new input, does it mean that I have to modify my model to take 1X1024 sized input?"
> The ELMo representations in our paper are 1024 dimensional -- 512 for both the forward and backward LSTMs.  If your existing input is size 512 then after adding ELMo the new input dimension will be 512 + 1024.
>
> "By task model do you mean the biLM?"
> In our paper, the "task model" refers to the task specific supervised NLP model, e.g. the entailment model, or Q&A model, etc.  In your case it is the NMT system.  The biLM is the unsupervised pretrained component.
>
> "In either case, could you elaborate more on "including ELMo at the output of task RNN"?"
> In our paper, the task RNN's input is GloVe (or other pretrained emeddings) concatenated with a layer of ELMo representation, [x_k; ELMo_k].  It outputs hidden states h_k for each token.  In some cases (e.g. SQuAD, SNLI) including another layer of ELMo representations at the output layer also helped.  This is accomplished by replacing h_k with [h_k; ELMo_k].  In these cases, ELMo is included twice in the task model, and each location uses a different set of learned scalar weights.  See Section 3.3 for more details.

---

### Public Comment · (anonymous) · 2017-12-05
**Nice Contribution**

I think this paper shows something that was being speculated about in the past years. Finally, someone proves this on a wide variety of tasks. The results are convincing in my view and the contribution is important, especially if the authors would manage to open source this for a variety of DL frameworks.

Good work.

---

> ### Author Response · Authors · 2017-12-07
> **Glad you enjoyed the paper!**
>
> Thanks for your comment and glad you enjoyed the paper!  We plan to release both tensorflow and pytorch implementations.

---

### Public Comment · (anonymous) · 2018-01-22
**Word level BiLM**

Nice work! Would you happen to know how ELMo does with purely word-level inputs? In other words, if only a word-level BiLM was pre-trained? Do any of the baselines use character-level or subword features? Maybe there's not enough data in each task to train a character level model end-to-end and that's part of the perf increase seen here. It would be interesting to separate the performance increase from subword information + contextual information.

---

> ### Author Response · Authors · 2018-01-23
> **Character vs contextual information**
>
> Thanks for the question!
>
> Some of the task models (SQuAD, coref, NER) include character CNN layers, the others to not.
>
> We haven't done a careful comparison of a word level biLM to the character based one in the paper.  However, we have other evidence that shows most of the performance increase is from the biLM's contextual vs subword information.
>
> First, the learned weights for all the task models focus attention on the contextual biLSTM layers from the biLM over the token layer (Section 5.5).
>
> Second, in some additional follow-up experiments, we compared the context independent biLM token layer ($h_k^{LM,0}$ in Section 3.2) to pre-trained GloVe (Pennington et al 2014)and fastText vectors (Bojanowski et al 2017) in the SQuAD model.  The biLM token layer includes 2048 character CNN filters, two highway layers and a feedforward layer but does not include any contextual information.  The fastText vectors include subword information and GloVe does not.
>
> SQuAD development set F1
> -------------------------------------
> With GloVe:                           80.8
> With biLM token layer:        81.4
> With fastText:                       81.6
> With ELMo:                           85.6
>
> From the table, most of the improvement with ELMo is from the contextual information with a 4.8% increase over the GloVe baseline, compared to increases of 0.6% and 0.8% with adding subword information via the biLM token layer or fastText.

---

### Author Response · Authors · 2018-02-03
**NLP conference**

As suggested by the meta reviewer, the empirical results in this paper are better suited for a NLP conference then ICLR.  As a result, we are withdrawing it from ICLR.  Thank you to the Chairs for your consideration and paper acceptance.

---

### Decision · Program_Chairs · 2018-01-29
**ICLR 2018 Conference Acceptance Decision**

**Decision:**

Accept (Poster)

**Comment:**

This is a good paper that presents state-of-the-art results on a number of challenging NLP tasks. The idea is fairly simple and clean, I therefore expect it to get adopted in the community. It also seems to work across several tasks, which is nice. At the same time it is fairly simple (train an LSTM language model and use representations from all levels of the LSTM in the input or output layer of a supervised task of interest) and hence may be more appropriate for an NLP conference than ICLR?

It is a good paper and it will get cited even though the ML contributions are modest. Accept due to the strong empirical results.